# Whole Genome Resequencing of 205 Avocado Trees Unveils the Genomic Patterns of Racial Divergence in the Americas

**DOI:** 10.3390/ijms262110353

**Published:** 2025-10-24

**Authors:** Gloria P. Cañas-Gutiérrez, Felipe López-Hernández, Andrés J. Cortés

**Affiliations:** 1Corporación Colombiana de Investigación Agropecuaria (AGROSAVIA), I La Selva, Km 7 vía Rionegro-Las Palmas, Rionegro 054048, Colombia; llopez@agrosavia.co; 2Facultad de Ciencias Agrarias—Departamento de Ciencias Forestales, Universidad Nacional de Colombia—Sede Medellín, Medellín 050034, Colombia; ancortesv@unal.edu.co

**Keywords:** *Persea americana* L., germplasm bank, ‘Plus Tree’, ex situ conservation, northwest South America, low-coverage whole genome resequencing (lcWGS)

## Abstract

Avocado (*Persea americana* Mill.) is one of the most widely consumed fruits worldwide. The tree species is traditionally classified into three botanical races: Mexican, Guatemalan, and West Indian (with a potentially distinct Colombian genepool). However, previous studies using molecular markers, such as AFLPs, microsatellites (SSRs), and GBS-derived SNP markers, have only partially resolved this racial divergence, especially in the hyper agrobiodiverse region of northwest South America. Therefore, in order to confirm genetic identity and origin of “criollo” avocado cultivars in the region, as well as to improve their traceability as rootstocks for the Hass variety, we performed low-coverage whole genome resequencing (lcWGS) on 205 ex situ conserved tree samples, comprising 42 commercial varieties and 163 “criollo” trees from various provinces in Colombia. This characterization yielded a total of 64,310,961 SNPs at an average coverage of 4.69×. Population structure analysis using principal component analysis (PCA) and ADMIXTURE retrieved at least five genetic clusters (*K* = 5), partly confirmed by Bayesian phylogenetic inference. Three clusters matched the recognized Mesoamerican botanical races (Mexican, Guatemalan, and West Indian), and two clusters reinforced the distinctness of two novel Andean and Caribbean Colombian genetic groups. Finally, in order to retrieve high-quality SNP markers for racial screening, a second genomic dataset was filtered, consisting of 68 avocado tree samples exhibiting more than 80% ancestry to a given racial cluster, and 9826 SNPs with a minimum allele frequency (*maf*) of 5%, a minimum sequencing depth (SD) of 10× per position, and missing data per variant not exceeding 20% (i.e., variants with genotypes present in at least 80% of the samples). This racially segregating high-quality subset was analyzed against the racial substructure using linear mixed models (LMMs), enabling the identification of 254 SNP markers associated with the five avocado genetic races. The previous candidate SNPs may be leveraged by nurseries and producers through a high-throughput SNP screening system for the racial traceability of seedling donor trees, saplings, and rootstocks. These genomic resources will support the selection of regionally adapted elite rootstocks and represent a landmark in Colombian horticulture as the first large-scale lcWGS-based characterization of a local avocado germplasm collection.

## 1. Introduction

Avocado (*Persea americana* Mill.), a subtropical tree that belongs to the Lauraceae family, is a fruit crop known for its high nutritional value and economic importance [1]. Rich in monounsaturated fats and various vitamins, the avocado’s global popularity has been increasing over the years due to the rising demand for healthy foods and alternative dietary fat sources [2]. In parallel to this trend, world avocado production has grown rapidly in recent years, increasing from an annual global production of 4.07 million tons in 2011 to 8.69 million tons in 2021. In the same period, Latin America contributed significantly to global avocado production, with Mexico, Colombia, and Peru adding 1.68 and 4.20 million tons in 2011 and 2021, respectively (as per 2022 FAO). In 2022, the avocado market was estimated at USD 14.55 billion globally and is projected to reach USD 26.04 billion by 2030. However, grafted avocado plantations are typically monoclonal for the Hass scion clone, while relying on highly heterogenous seedling rootstocks with poor traceability from the seedling donor tree [3]. Therefore, current avocado pre-breeding strategies are expanding germplasm characterizations to identify and utilize locally adapted rootstocks resistant to *Phytophthora cinnamomi* [4,5,6,7], while procuring, at the same time, diversifying the repertoire of commercially accepted clonal scions [8].

Avocados have typically been classified into three horticultural races [9]: Mexican, *P. americana* var. *drymifolia* (Schlecht. et Cham.) Blake; Guatemalan, *P. americana* var. *guatemalensis* (L.) Wms.; and West Indian, *P. Americana* var. *americana* Mill. [10]. The Mexican race, from the mid-altitude highlands in Mexico, is recognized by its early fruit maturity and cold tolerance. Meanwhile, the Guatemalan race, from the mid-altitude highlands of Guatemala, is known for having small fruits and late fruit maturity. On the other hand, the West Indian race, from the lowlands in southern Mexico and Central America, is characterized by larger fruits with low oil content [10]. Most commercial cultivars are regarded as hybrids of these three races, mainly crosses between the Guatemalan and the Mexican types like the Hass cultivar [11]. This racial stratification has been validated by modern genetic markers like single sequence repeats (SSRs) [11,12,13,14,15,16,17,18] and single-nucleotide polymorphism (SNPs) [10,19,20,21,22,23], suggesting more genetic proximity between the Mexican and Guatemalan races. More recently, GBS-derived SNP markers have suggested a fourth Colombian race in northwest South America, dating from the Pleistocene before human colonization, and more closely related to the lowland West Indian race [24].

Despite these previous achievements, and the nutritional and economic importance of avocado in tropical and subtropical regions, significant gaps remain in understanding the population structure and racial ancestry of local avocado cultivars in northwest South America, a putative convergence area of multiple races, and home to high levels of agrobiodiversity [25]. In particular, the genetic distinctiveness, origin, and traceability of “criollo” avocados in the region remain poorly defined, limiting their integration into commercial breeding and rootstock selection programs. Therefore, this study involved resolving these issues by using low-coverage whole-genome resequencing (lcWGS). Specifically, it aims to (*i*) elucidate the genomic patterns of population stratification in the Colombian avocado germplasm, and (*ii*) identify race-informative SNP markers for practical applications in cultivar discrimination and traceability. We hypothesize that lcWGS would corroborate that Colombian “criollo” avocados harbor unique genomic lineages beyond the three classical botanical races [24], and that these lineages can be reliably differentiated using high-quality SNP markers. This work not only refines our understanding of avocado racial divergence in a key center of diversification, but also provides genomic tools to improve cultivar authentication, guide rootstock selection, and promote the conservation and valorization of native genetic resources.

## 2. Results

To explore the population structure and racial ancestry of the avocado germplasm from northwest South America, we conducted low-coverage whole-genome resequencing (lcWGS) on 205 avocado tree samples, including 163 regionally conserved “criollo” trees and 42 commercial varieties. This screening led to over 64 million SNPs, later imputed into robust phylogenetic and population structure inferences. Our results confirmed the three classical Mesoamerican botanical races, Mexican, Guatemalan, and West Indian, and corroborated two additional, genetically unique clusters that are exclusive to Colombia’s Andean and Caribbean regions. We further identified a subset of high-confidence SNPs from racially pure individuals and applied linear mixed models (LMMs) to detect ancestry-informative markers. These findings provide new insights into avocado racial divergence and offer scalable genomic tools for racial traceability and rootstock selection.

### 2.1. lcWGS Recovered More than 64 Millon SNPs

Whole-genome resequencing (lcWGS) of 205 avocado tree samples (Appendix A), comprising both Colombian “criollo” trees and commercial varieties, yielded on average 29,718,869 reads and 4,487,549,221 bp per sample, with a mean G:C content of 39.97%, and Q20 and Q30 of 97.60% and 93.49% (Appendix A). On average, 28,735,593 reads (96.69%) were retained after applying quality filters, with a mean length of 4,280,892,756 bp and an acceptable [26] mean sequence depth of 4.69× per sample (Appendix A). A total of 28,586,409 (99.48%, duplicates rate of 11.55%) reads per sample mapped against the Hass avocado reference genome (Appendix A), 81.29% at a mean minimum depth of 1×, and 44.74%, 4.88%, and 1.26% at mean minimum depths of 4×, 10×, and 20× (Appendix A).

SNP calling on this dataset yielded an average transition/transversion (Ts/Tv) ratio of 2.19 (Appendix A) and a total of 64,310,961 SNPs (Figure 1, Appendix A), with mapping quality (MQ) ≥ 40 and quality depth (QD) ≥ 2. Annotation showed that a large fraction of SNPs (86.67%, Appendix A) mapped to non-coding regions (63.74% intergenic and 22.93% intronic), though we also detected numerous coding (2.9%; 1.16% synonymous, 1.68% nonsynonymous, and 0.05% and 0.01% causing a gain and a loss of a stop codon) and flanking variants (8.4%; 4.48%, and 3.76% 1 kb upstream/downstream, a transcription start/end site).

### 2.2. Phylogenetic and Population Structure Analyses Reveal Five Genetic Clusters

Using the full panel of 205 trees, a Bayesian phylogeny (on SNPs with *maf* ≥ 5%, depth ≥ 10×, missing data ≤ 20%) partly grouped samples into five non-monophyletic clusters (Figure 2), three classical botanical races, Mexican (ME), Guatemalan (GU), and West Indian (WI) races, and two distinct Colombian groups: Andean Colombian (CoA), mostly from the Andean Antioquia province, and Caribbean Colombian (CoCA).

Principal component analysis (PCA) on the entire dataset further confirmed the distinctiveness of these clusters (Figure 3, Appendix A), with PC1 and PC2 accounting for 21.69% and 3.96% of variance and clearly separating the five groups.

ADMIXTURE software in the same full panel retrieved the lowest cross-validation error at *K* = 5 (Appendix A), reinforcing the presence of five genetic ancestries across the 205 avocado samples (Figure 4), without *K* = 6 (Appendix A) offering additional discrimination. Notably, most “criollo” trees from Colombia showed high ancestry proportions (>80%) to either the CoA or CoCA clusters (Appendix A), suggesting limited introgression from Mesoamerican races.

Colombian samples showed geographically coherent clustering in both PCA and admixture plots. Andean samples, mainly from Antioquia, clustered separately from Caribbean samples that originated from lowland provinces. This geographic stratification aligned with the population structure, suggesting that Colombian “criollo” avocado trees have experienced long-term geographic or ecological isolation. Their genetic distinctiveness was comparable to that observed between Mexican and Guatemalan races, indicating that the Colombian germplasm may be a secondary center of avocado diversification.

### 2.3. LMMs Recovered 254 Race-Informative SNP Markers

To identify markers suitable for racial tracing, we applied additional quality filters (*maf* ≥ 5%, depth ≥ 10×, missing data ≤ 20%) to extract a high-confidence SNP subset. After this filtering, a total of 9826 SNPs were retained (Appendix A), depicting the same demographic signatures as the full dataset (Appendix A). Samples with >80% ancestry (as per [27]) to a single genetic cluster were kept (*n* = 68) to avoid admixture-related signals. PCA on this racially pure subset retrieved the five genetic clusters (Figure 5A, Appendix A), confirming the suitability of the subset for association analyses targeting racial stratification.

Linear mixed models (LMMs) applied to the 68 racially pure samples identified 254 SNPs (Appendix A) associated with genetic race (strict Bonferroni threshold of *−log_10_* = 5.29). These SNPs showed high discriminatory power in the PCA (Figure 5B, Appendix A) and Bayesian inference (Appendix A), successfully recapitulating the five-cluster genetic structure, this time with minimal overlap. Most of these SNPs mapped to non-coding regulatory regions, as expected for putatively neutral, demographic-informative markers, not necessarily in linkage disequilibrium (LD) with selection footprints. These markers are candidates for integration into high-throughput genotyping platforms for racial identity screening in nurseries, aiding in seedling rootstock traceability and elite donor selection.

## 3. Discussion

The current study represents the most comprehensive genomic characterization to date of the avocado germplasm, leveraging lcWGS to resolve long-standing ambiguities in racial classification and genetic ancestry, especially in the hyper agrobiodiverse region of northwest South America [28]. We retrieved over 64 million SNPs in 205 avocado trees. After inputting this impressive dataset in robust population structure and phylogenetic analyses, we confirmed the classical Mexican, Guatemalan, and West Indian botanical races, while also corroborating two genetic clusters unique to Colombia’s Andean and Caribbean regions. These findings refine our understanding of avocado’s racial diversity, challenging traditional race boundaries, and reinforcing Colombia’s role as a secondary center of avocado diversification. Additionally, the generation of 254 ancestry-informative SNPs provides a scalable marker chip for racial traceability and nursery-level selection, harnessing genomic-enabled applications for rootstock breeding and avocado “criollo” tree conservation strategies. These novel genomic resources for avocado will certainly assist additional efforts in germplasm screening for standing adaptation signatures, genetic mapping of key agronomic and resistant traits with the potential to be leveraged through rootstock selection, and predictive breeding for seedling rootstock improvement.

### 3.1. lcWGS Corroborates Genomic Racial Stratification and Two Colombian Avocado Clusters

This work has demonstrated that the application of lcWGS enables a comprehensive and high-resolution reconstruction of the genomic landscape of the understudied Colombian avocado germplasm. Over 64 million SNPs across 205 samples recovered a clear genetic structure that aligns with the three recognized botanical races (Mexican, Guatemalan, and West Indian) [29], but also verified two additional genetic clusters exclusive to the Colombian “criollo” avocados. These clusters, corresponding to distinct Andean and Caribbean groups, were partly supported by both ADMIXTURE software and Bayesian phylogenetic inferences. The full-genome evidence confirms and expands upon earlier hypotheses based on GBS-derived SNPs [24], reinforcing the presence of regionally differentiated lineages in the hyper agrobiodiverse region of northwest South America [28], likely shaped by isolation by distance (IBD) and animal dispersion dating to the Pleistocene, rather than more recent human-mediated dispersal and cultural selection [30]. These lcWGS results emphasize the need to revise traditional racial classifications [8], envisioning the inclusion of Colombia as a main center of avocado diversification.

The genetic distinctiveness of these Colombian lineages is further supported by their geographic coherence and ancestry proportions. Samples from the Andean range and Caribbean lowlands exhibited high membership coefficients to their respective clusters, suggesting limited introgression from the classical Mesoamerican races. This pattern is insightful given Colombia’s geographic location at the junction of Central and South American floras [28], where hybridization and gene flow has been observed to blur the genepool and racial boundaries in other American crops [31,32,33]. Instead, the genetic consistency of these clusters may reflect geographic and ecological isolation shaped by the topographic and environmental complex heterogeneity in the northern Andes and the Caribbean plains [34,35]. The differentiation of the Colombian clusters may as well be due to recent divergent selection pathways, consistent with the region’s status as a hotspot of agrobiodiversity [25], and independent avocado management by Indigenous and campesino communities [29]. Data not only reinforces prior insights from lower-resolution marker genotyping systems such as GBS, but also highlights how genome-wide data can resolve fine-scale differentiation in highly heterozygous perennial tree crop species [20].

The recognition of distinct Colombian avocado lineages also has key consequences for conservation and future breeding. As worldwide demand for avocado continues to rise, and young plantations are being established all over the South American, African, and Asian tropics and sub-tropics, the genetic base of commercial production remains narrow and heavily reliant on the Hass-like cultivars and the few selected rootstocks (e.g., Duke-7) [3]. The lcWGS genomic characterization reported here highlights the potential of the native Colombian germplasm as a likely reservoir of novel alleles for disease resistance [36], climate resilience [37], sustainable nutrient-use efficiency [38,39,40,41,42], and fruit quality traits [43]. A first step to unlock and utilize this potential is through the selection of seedling donor trees for rootstock propagation during grafting [17], targeting resistance to *P. cinnamomi*, adaptation to local conditions, and water use efficiency. A second mid-term inevitable step would be to scale this multi-trait selection scheme to an optimized and diverse pool of clonal elite rootstocks by standardizing layering [44], double grafting [45], and micro-cloning [46] techniques. A third long-term path for utilizing these hidden pockets of cryptic avocado diversity would involve revaluing and boosting marketing strategies for local avocados to be used as scions, despite the fact that they tend to resemble the West Indian fruit type more than the commercially accepted Mexican × Guatemalan hybrids [18]. Overall, validating two Colombian clusters reinforces the importance of ex situ conservation programs, and supports further integration of local avocado diversity into rootstock (and even scion) breeding schemes. By doing so, avocado agrobiodiversity could contribute to a more resilient and sustainable avocado industry, while also promoting the biocultural heritage embedded in Colombia’s criollo varieties.

### 3.2. Candidate SNPs for Racial Tracing Supports Diversified Avocado Rootstock Production

In addition to confirming the existence of unique Colombian avocado lineages, this study further provides practical tools for racial identification, traceability, and downstream applications in nurseries and racial-focus horticultural management systems. Through stringent filtering and LMMs, we identified 254 high-quality SNPs associated with the five genetic clusters, including those specific to the Andean and Caribbean “criollo” groups. These SNPs are candidates to be escalated through cost-effective high-throughput genotyping assays (e.g., KASPar [47] and SNP chips [48,49,50], successfully applied in other perennials [51,52]) to trace the racial ancestry of individual seedling donor trees, saplings, and rootstocks. This, in turn, has direct implications for nurseries and producers seeking to enhance seedling traceability, select regionally adapted rootstocks, and ensure genetic consistency in grafting practices [49,50].

The availability of ancestry-informative markers helps overcome one of the persistent challenges in avocado breeding, which is the undocumented origin and high heterozygosity of seedling rootstocks [20], both factors having traditionally bottlenecked selection efforts. This panel of candidate SNPs would enable nurseries to enhance the value of their avocado grafting material by pre-screening seedling rootstocks for racial identity prior to grafting [49,50]. Producers would gain access to a broader and more customized offer of planting material: (*i*) seedling rootstocks with full traceability to the donor tree, (*ii*) seedling rootstocks pre-labeled by racial background, and (*iii*) elite clonal rootstocks [45,46].

These options differ in terms of genetic uniformity and cost, letting growers make informed choices based on their budget constraints and target markets [53]. Diversifying the supply of avocado planting material will not only support the establishment of new orchards with improved resilience, sustainability, and fruit quality, but also promote greater social inclusion by empowering smallholder farmers and medium-scale producers in an industry that is traditionally dominated by large-investment entrepreneurs.

### 3.3. Perspectives

A much-expected imminent next step following the genomic confirmation of avocado racial stratification, including the two Colombian clusters, is to unveil the genomic architecture of divergence among races with the available lcWGS data. Specifically, the identification and comparison of genomic islands of divergence [54,55,56] would serve as independent validation to determine if similar genomic regions have been repeatedly recruited during the diversification of the Mexican, Guatemalan, West Indian, and Colombian lineages, or whether different loci underlie their independent evolutionary trajectories. Such analyses could clarify whether avocado races share common targets of selection or reflect lineage-specific divergence. Alternatively, genomic divergence among races may fortuitously be shaped by genomic constraints, such as translocations or recombination and mutation rate variation [57], all of which are known to make genetic drift and incomplete linage sorting more likely, due to the reduced effective population size at specific locations in the genome [54,58]. Still, reconstructing the genomic islands of divergence and disentangling the underlying selective, demographic, molecular, and spurious drivers would offer insights into the evolutionary dynamics of perennial fruit tree crops.

In parallel, future research should also pursue lcWGS-based genome-environment association (GEA) studies across the avocado distribution range to uncover the standing genomic basis of ecological adaptation [59,60,61,62]. GEA approaches can identify loci associated with climate adaptation from the highlands of Mexico and Guatemala, e.g., [63], where the Mexican and Guatemalan races evolved, to the lowlands of Central America and northwest South America, where the West Indian and Colombian lineages are found. Some of these signatures of adaptation may coincide with specific genomic islands of divergence, particularly when the latter are shaped by environmentally driven selective sweeps [64]. Compiling GEA results across a wide spectrum of bioclimatic variables and eco-physiological indices [65,66,67,68], and contrasting them against the among-races genomic divergence profiles, could inform regional avocado breeding strategies for the selection of climate-resilient rootstocks targeting local environmental conditions.

Another promising research avenue is to utilize the lcWGS data to identify candidate genes for biotic stress tolerance, particularly for *P. cinnamomi*, a soil-borne fungal pathogen responsible for avocado root rot [69]. Given the cryptic genetic diversity within the Colombian “criollo” avocado clusters, comparative analyses of resistance gene complexes across the identified populations could help with forecasting the natural resistance potential in currently uncharacterized avocado trees. This may speed up the identification and propagation of disease-resistant rootstocks without exclusively relying on more imprecise inoculation-based screening methods [4,6,7,70], thereby enhancing orchard resilience.

Finally, future work should aim to reconstruct the genetic basis of key agronomic traits that are likely to be modulated by rootstocks [71,72,73,74]. As per previous rootstock-mediated heritability estimates [17,18], phenotypic variation influenced by the rootstock genotype could span from environmental adaptation and water-use efficiency [75,76,77], to overall productivity and fruit quality [38,78,79,80,81], among other traits [82,83]. lcWGS enables high-resolution whole genome association studies (WGAS) [84,85], narrowly mapping quantitative trait nucleotides (QTNs) across genetic and environmental backgrounds [86].

Once such candidate trait-associated makers are independently validated, they could then be integrated with the 254 SNPs suggested here for racial tracing, enabling indirect marker-assisted (MAS) and genomic (GS) selection pipelines [84,87,88,89]. Both approaches could in turn facilitate greater efficiency and precision during an early pre-screening of saplings and seedling rootstocks at nurseries [90], not just for racial identity but also for agronomic trait prediction [49,50,91] and bioactive compound identification [92]. These efforts will advance the integration of genomics into rootstock selection schemes, supporting the transition to more diversified and sustainable avocado production systems.

## 4. Materials and Methods

### 4.1. Plant Material

Fresh young leaf samples were collected from 205 avocado trees from two sources: 43 “criollo” avocado trees sampled in the province of Antioquia, northwest Andes of Colombia, 39 of which are conserved ex situ at AGROSAVIA’s research station La Selva in Rionegro (Antioquia province), and 162 avocado samples belonging to the avocado collection of the Colombian Germplasm Bank (CGB) located in Palmira (Valle del Cauca), of which 120 were “criollo” avocado trees from different provinces in Colombia and 42 corresponded to commercial varieties (Table 1). Leaf samples were preserved in moisture-absorbing, color-indicating silica gel (Sigma Aldrich, Taufkirchen, Germany).

### 4.2. Library Preparation and Genome Sequencing (lcWGS)

Genomic DNA of foliar tissue from the 205 avocado samples was extracted with the DNeasy Plant Mini Kit (QIAGEN, Hilden, Germany). DNA purity and integrity was verified by Agarose gel electrophoresis analysis. DNA purity was determined by evaluating the OD260/OD280 ratio, with measurements performed using a NanoDrop^®^2000 spectrophotometer (Thermo Fisher Scientific, Waltham, MA, USA). PicoGreen and Qubit quantitation were also carried out for accurate measurement of DNA concentration. DNA samples with an OD260/OD280 ratio between 1.8 and 2.0, and a total amount more than 0.5 µg, qualified for library preparation. Genomic libraries were constructed with Illumina TruSeq™ Nano DNA Sample Prep Kit (Illumina, San Diego, CA, USA). The samples were sequenced using high-throughput DNA sequencing, specifically, pair-end sequencing on the Illumina Novaseq platform, with a read length of 150 bp at each end. Library preparation and sequencing were performed at CD Genomics (South Plainfield, NJ, USA).

### 4.3. Raw Sequencing, Data Proccessing, and Quality Control Statistics

The original high-throughput sequencing data recorded in image files was first transformed into sequence reads by base calling with the CASAVA software (v.1.8.2) (Illumina Inc., San Diego, CA, USA). Sequences and sequencing quality information were stored in FASTQ files. Adapter contamination and low-quality reads were removed from the raw data to obtain clean reads for subsequent analyses (Appendix A).

Specifically, data preprocessing was carried out using the sliding window method implemented in *fastp* (v0.20.0), as follows: (*i*) removal of 3′ end adapter contamination, (*ii*) quality filtering using a sliding window approach with a window size of 5 bp, sliding from the 3′ end to the 5′ end, to compute the average *Q* value of bases within the window (so that if the *Q* value < 20, bases within the window were removed; otherwise if the *Q* value ≥ 20, the sliding process was halted), (*iii*) length filtering by discarding paired-end sequences < 50 bp, and (*iv*) N-base filtering by eliminating paired-end sequences, for which the number of N bases was ≥5.

### 4.4. Mapping to Reference Genome

The genome of the Hass avocado was used as a reference (https://www.avocado.uma.es/easy_gdb/downloads.php accessed on 13 October 2025). The preprocessed data were mapped to the reference genome, using BWA (0.7.12-r1039) with default parameters. The SAM files were sorted and converted to BAM files using Picard 1.107 software (http://www.psc.edu/index.php/user-resources/software/picard accessed on 13 October 2025). The “FixMateInformation” command was employed to ensure consistency in all paired-end reads’ information. The Picard software package was further used to remove duplicates with the “MarkDuplicates” tool, meaning that if multiple paired reads mapped to the same chromosomal coordinates, only the one with the highest score was retained. This was done to account for processing and optical duplicates, which, if retained, could be misinterpreted as evidence for variant detection.

### 4.5. SNP Detection, Distribution, and Annotation

The detection of SNPs was performed using the GATK software v.4.5.0.0 [93]. The specific steps were as follows: (*i*) the “Haplotype Caller” module of GATK was used to identify variant sites, generating GVCF files for each sample. Subsequently, (*ii*) the “Combine GVCFs” module was used to merge all sample GVCF files by chromosome, and then (*iii*) the “GatherVcfs” module was used to merge all chromosome GVCF files to obtain a GVCF file for all samples. Finally, (*iv*) the “Select Variants” module was used to extract SNPs and INDELs, and (*v*) the “Variant Filtration” module was used for quality control filtering to obtain the final SNP and INDEL files.

To ensure the reliability of SNP sites, further filtering of the obtained SNP sites was performed according to the following criteria: (*i*) Fisher test of strand bias (FS) ≤ 60, (*ii*) mapping quality (MQ) ≥ 40, (*iii*) quality depth (QD) ≥ 2, (*iv*) ReadPosRankSum ≥ −8.0 and (*v*) MQRankSum > −12.5. After filtering according to the above conditions, the number of group SNPs obtained in each sample was statistically recorded. SNP annotation was done using ANNOVAR software (v.2024-08-16) [94], and visualization used 0.1 Mb sliding windows. These steps led to a total of 64,310,961 SNPs for all 205 avocado samples.

### 4.6. Phylogenetic Tree Reconstruction Based on lcWGS-Derived SNPs

To infer the evolutionary relationships among the analyzed avocado samples, a Bayesian phylogenetic analysis on SNPs with *maf* ≥ 5%, depth ≥ 10× and missing data ≤ 20% was conducted in MrBayes v.3.2.7a, under a controlled Conda environment for computational reproducibility. Two independent runs were executed with the GTR+Γ substitution model, each with two Markov chains for 2 × 10^6^ generations, a chain temperature of 0.9, sampling every 1000 generations, and discarding the first 25% as burn-in. Convergence was assessed by monitoring the average standard deviation of split frequencies (ASDSF < 0.03). Consensus trees were generated in the *sumt* command with the *allcompat* criterion and later visualized and edited in Geneious Prime v.R.9.1.8. Racial controls followed the morpho-agronomic classification previously compiled by Berdugo et al. [24].

### 4.7. Genetic Structure of Diverse Avocado Tree Samples

Unsupervised clustering via principal component analysis (PCA) was performed using the full dataset in order to analyze a simplified version of the dataset by reducing its dimensionality while retaining the features that contribute most to its variance. The analysis excluded SNPs with minor allele frequency (*maf*) below 0.05. The PCA was performed using the GCTA software (v.1.94.1) (https://yanglab.westlake.edu.cn/software/gcta/ accessed on 13 October 2025) and visualized in R (v.1.4.0).

To further understand the distribution of genetic variation among “criollo” avocado trees, an unsupervised population admixture genetic structure analysis was performed on the same dataset from the 205 avocado samples. For that, the genetic stratification of populations was analyzed using the ADMIXTURE v.1.3.0 software (http://dalexander.github.io/admixture accessed on 13 October 2025). The analysis involved setting *K* = 2–10 (assuming the existence of 2–10 ancestral populations) with model selection using the admixture model, while the remaining parameters were kept at their default settings. The value of *K* that approximated the true structure was determined based on the cross-validation (CV) error values for different *K* values. These population structure analyses allowed assessing and quantifying ancestry for each of the 205 avocado samples, distributed in five genetic clusters: ME, GU, WI, and CO, the latter of which was composed of two subgroups: Andean Colombian (CoA) and Caribbean Colombian (CoCA).

### 4.8. Identification of SNPs Associated with Avocado Races

From the VCF file with 64,310,961 SNPs obtained for the 205 avocado samples, a second round of filtering was carried out using the VCFtools software (v.0.1.16) [95]. SNP markers were retrained if they had a minimum allele frequency (*maf*) of 5%, a minimum of 10X of sequencing depth (SD) per position, and missing data per variant not exceeding 20% (i.e., variants with genotypes present in at least 80% of the samples). With this second SNP dataset, avocado samples exhibiting more than 80% ancestry (following [27]), as determined by the results of the population admixture analysis, were kept for downstream analyses. According to the filtering parameters described above, the second genomic dataset consisted of 68 racially pure avocado samples and 9826 high-quality SNP markers.

The population structure in this simplified, more racially segregating, dataset was inspected as per a principal component analysis (PCA), performed in the TASSEL software v.5 [96]. The very same dataset and software were also utilized to compute linear mixed models (LMMs), which used a kinship relationship matrix as random effect, aiming to link SNP marker variation with population stratification in the five avocado races (coded as ME, GU, WI, CoA, and CoCA). Finally, the discriminatory power of the SNPs associated with the racial stratification, at a strict Bonferroni threshold of *−log_10_* = 5.29, was reexamined via a PCA in the same software but only retaining the associated SNPs.

## 5. Conclusions

This work addressed the racial genomic divergence of avocado trees in one of its most agrobiodiverse yet understudied regions, northwestern South America. Screening via lcWGS a total of 205 avocado tree samples, including local “criollo” trees and commercial cultivars, enabled confirming that the Colombian avocado germplasm harbors distinct genetic lineages [24] that are genetically divergent from the three traditionally recognized Mesoamerican botanical races (i.e., Mexican, Guatemalan, and West Indian) [97]. Population genomic analyses validated the existence of two clearly differentiated Colombian clusters [34,35], corresponding to the Andean and Caribbean regions. Therefore, Colombia may represent a major center of diversification for avocado, likely shaped by isolation by distance (IBD), and topographic and ecological heterogeneity [34,35].

This study also succeeded in generating novel genomic resources to enable racial traceability and improve the selection and propagation of avocado rootstocks [17,18]. From a subset of 9826 high-quality lcWGS-derived SNP markers, we identified 254 candidate SNPs that were significantly associated with genetic race. This marker panel is scalable, as a genomic forecasting tool to empower nurseries and producers who may be interested in pre-screening seedling donor trees, saplings, and rootstocks for racial identity, thereby improving traceability and enabling more uniform grafts [49,50].

These findings reinforce the need to revise current racial classification frameworks in avocados, while formally incorporating the Colombian clusters into a broader taxonomy and ex situ conservation planning, e.g., [98,99]. The identification of distinct Colombian lineages also highlights the untapped potential of the local germplasm as source of unexplored alleles for root rot resistance, abiotic stress tolerance, and fruit quality. key traits in the global context of climate change and increasing avocado demand. The lcWGS genomic resources generated as part of this work are a milestone in bridging genomics with applied molecular breeding, ultimately enabling more traceable and locally adapted avocado production systems, while promoting the conservation and valorization of regional avocado biocultural heritage.

## Figures and Tables

**Figure 1 ijms-26-10353-f001:**
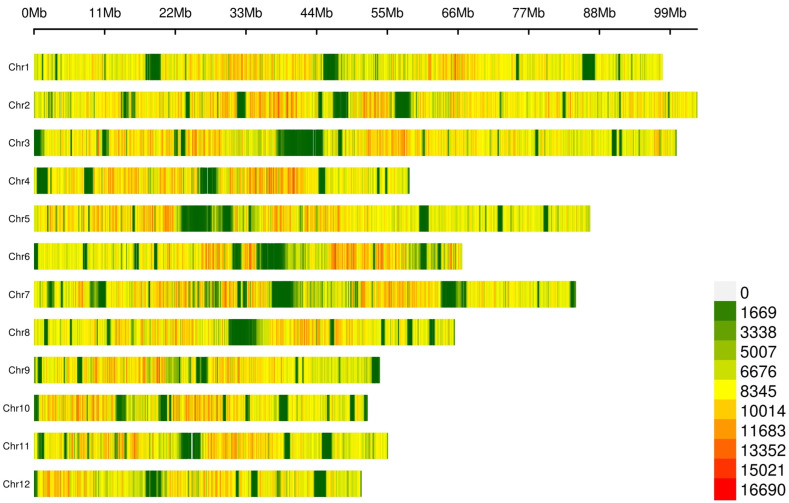
Distribution of lcWGS-derived SNP markers across the 12 avocado chromosomes. The number of SNPs was calculated through a 0.1 Mb sliding window.

**Figure 2 ijms-26-10353-f002:**
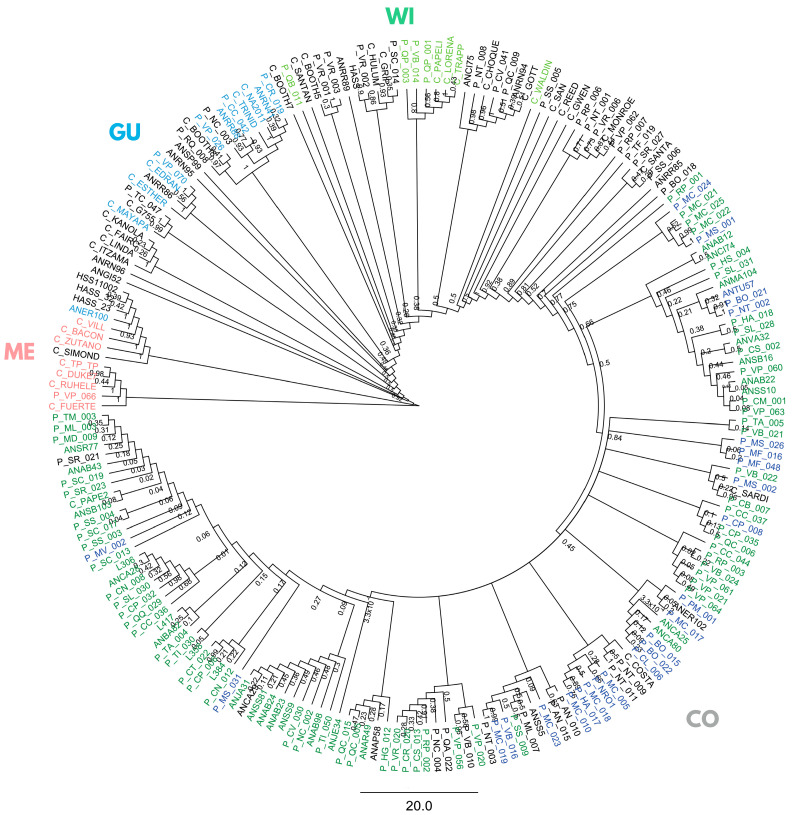
Bayesian phylogenetic inference in 205 avocado tree samples with lcWGS-derived SNP markers. Racial controls are colored in pink, light blue, and light green for Mexican (ME), Guatemalan (GU), and West Indian (WI) races, and dark blue and dark green for Caribbean (CoCA) and Andean (CoA) Colombian (CO). “Criollo” samples from the Antioquia province’s ‘Plus Tree’ collection in CI La Selva are named starting with “AN”, and those from the Arangro plant nursery start with “L”. “Criollo” samples from the Colombian Germplasm Bank (CGB) in CI Palmira are named starting with “P”, while commercial varieties and Hass controls start with “C” and “Hass”.

**Figure 3 ijms-26-10353-f003:**
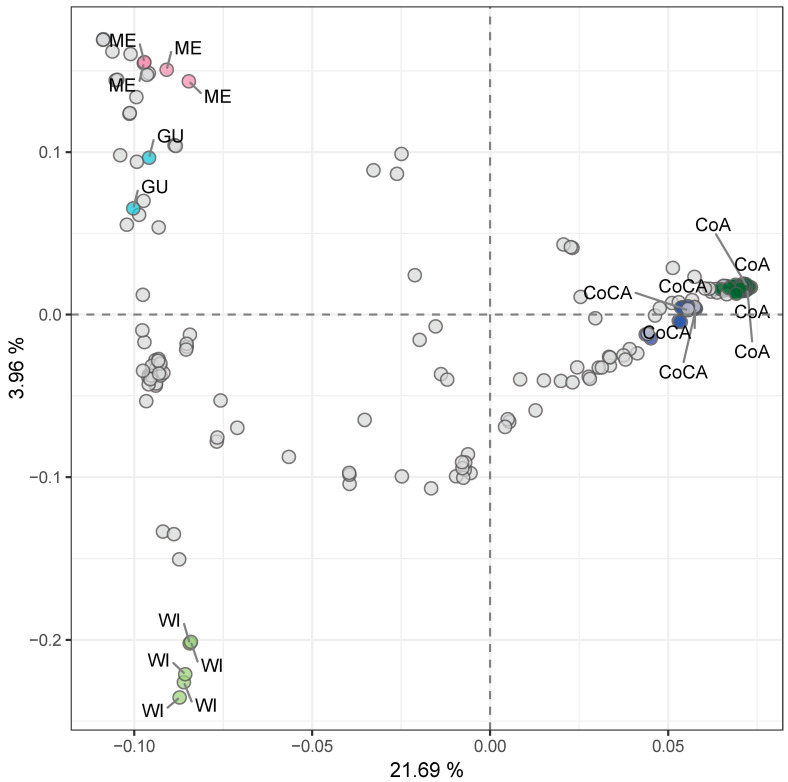
Unsupervised genetic clustering via principal component analysis (PCA) of 205 avocado tree samples with lcWGS-derived SNP markers. Racial controls are labeled and colored top down from left to right, as follows: pink for Mexican (ME) race, light blue for Guatemalan (GU) race, light green for West Indian (WI) race, dark blue for Caribbean Colombian (CoCA), and dark green for Andean Colombian (CoA). Explained variance by each component is marked in each axis label.

**Figure 4 ijms-26-10353-f004:**
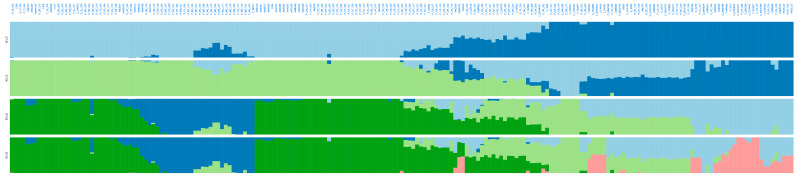
Admixture-based unsupervised genetic clustering (from *K* = 2 to *K* = 5) in 205 avocado tree samples with lcWGS-derived SNP markers. Lineages at *K* = 5 are colored as in Figure 3: dark green for Andean Colombian (CoA), dark blue for Caribbean Colombian (CoCA), light green for West Indian (WI) race, light blue for Guatemalan (GU) race, and pink for Mexican (ME) race.

**Figure 5 ijms-26-10353-f005:**
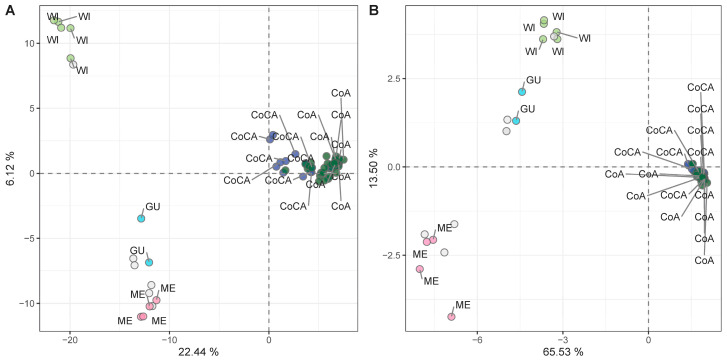
Principal component analysis (PCA) unsupervised genetic clustering in 68 racially pure (>80% ancestry to a genetic cluster) avocado tree samples with (**A**) 9826 high-quality SNP markers, and (**B**) 254 SNPs associated with racial classification (*K* = 5), as determined through LMMs.

**Table 1 ijms-26-10353-t001:** Summary of the 205 avocado trees sampled for low-coverage whole genome sequencing (lcWGS). NA stands for non-applicable and CI for research station. The full details are in Appendix A.

Type	Province	Ex Situ Collection	Total
“Criollo”	Antioquia	Avocado ‘Plus Tree’ Collection, CI La Selva	39
“Criollo”	Antioquia	Arangro Plant Nursery	4
“Criollo”	Antioquia	Colombian Germplasm Bank (CGB), CI Palmira	2
“Criollo”	Others	Colombian Germplasm Bank (CGB), CI Palmira	118
Commercial variety	NA	Colombian Germplasm Bank (CGB), CI Palmira	38
Avocado var. Hass	NA	Colombian Germplasm Bank (CGB), CI Palmira	4

## Data Availability

Processed data is contained within the article and as Appendix A. Raw data (205 FASTQ files) is available under NCBI BioProject ID PRJNA1311704.

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
