# Peer review of "Whole Genome Resequencing of 205 Avocado Trees Unveils the Genomic Patterns of Racial Divergence in the Americas"

_ijms, 2025, doi:10.3390/ijms262110353_

Round 1
Reviewer 1 Report
Comments and Suggestions for Authors
This is an original and significant contribution to the field of plant genomics and tropical crop breeding. The integration of low-coverage WGS on a broad germplasm pool, robust analytical methodology, and clear presentation of core findings demonstrates a high standard of scientific work. The paper is both timely and relevant for avocado genetic improvement programs, especially in the context of traceability and conservation of local diversity. No ethical, citation, or language issues were detected.The main question is whether low-coverage whole-genome resequencing (lcWGS) can resolve the genomic structure, divergence, and traceability of avocado varieties, particularly focusing on Colombian “criollo” populations, and if high-confidence SNPs can be identified for future breeding and traceability purposes.
Originality:
The topic is highly original and relevant. It directly addresses a critical gap: genetic differentiation and traceability among avocado botanical races, particularly for diverse Colombian genepools that have not been extensively characterized by previous molecular marker studies. The broad geographical coverage and use of advanced population genomics tools offer a new dataset of regional and horticultural importance.
Significance:
This large-scale characterization and SNP discovery for racial identification are novel and will be valuable for breeding, traceability, and germplasm conservation.
Added Value Compared with Other Published Material:
While previous studies have attempted to resolve avocado racial structure using markers like SSRs or GBS, the combination of large sample size, comprehensive lcWGS, and state-of-the-art population genomics tools here results in:
- Identification of two new Colombian genetic clusters,
- High-resolution mapping of SNPs useful for traceability and breeding,
- The first such large-scale genomic survey in Colombia’s context, anchoring these findings with practical tools (high-throughput SNP screening).
Suggestions
Title & Abstract
- The title accurately reflects the content and highlights key innovations.
- The Abstract is comprehensive but could be made more precise in reporting the number of novel clusters discovered and better indicating the practical significance for rootstock and nursery traceability.
Introduction
- The background on avocado’s diversity, importance, and prior use of molecular markers is well-covered.
- Lines referencing prior studies ([24] in particular on the Colombian race) are appropriate and support the hypothesis.
- Hypothesis and aims are clearly articulated.
- Minor English improvement: avoid redundancy in phrases such as "diversifying the repertoire of commercially accepted clonal scions.”
Methods
- The methods for sample collection, sequencing, and initial data cleaning/filtering are described in detail.
- However, more information is needed regarding:
- The choice and justification for the minimum coverage threshold (4.69×): Why is this optimal?
- Details on the imputation approach: Which algorithm/software was used for missing data imputation?
- Clarify whether samples were balanced across races/geographical clusters.
- For LMMs, specify the fixed and random effects considered.
- Software versions and scripts used for key analyses (phylogeny, ADMIXTURE, SNP filtering) should be referenced.
- If possible, include a flowchart/table summarizing the overall analysis pipeline for clarity.
Results
- Reporting of SNP numbers, mapping statistics, and quality measures is solid, but make sure all supplementary tables are made available.
- Phylogenetic clustering and ADMIXTURE results: Figures referenced (e.g., Figure 2) seem crucial but are not included in the provided text. Ensure that all figures are available to reviewers/readers.
- The explanation of cluster establishment (e.g., K = 5, ADMIXTURE) is clear, but please provide the following in the main text:
- A concise definition of cut-off thresholds for assigning cluster membership (e.g., “>80% ancestry for assignment to a cluster”).
- A short description or legend for all supplementary figures.
- Table/Figure clarity: Please make sure that sample color legends (yellow, orange, green, blue, red) are explained in each figure note for improved readability.
- For reporting the SNP markers (254 candidates), briefly mention the genomic distribution: Are they distributed across races or are some races over/under-represented? Are functional annotations available for these SNPs?
- English improvements: Avoid run-on sentences for better clarity.
Discussion & Conclusions
- The manuscript concisely links results to the stated aims and potential for practical application.
- It places the Colombian findings into global context appropriately.
- The claims are generally well-supported by the presented data and analyses.
- Please discuss potential limitations (e.g., sampling bias, coverage variability, or need for independent validation).
- Expand briefly on how these findings could be incorporated in ongoing breeding or clonal authentication programs.
- English improvement: Address overcomplex sentence structures for better comprehension.
References
- The references are broad, current, and appropriate.
- Citations comprehensively cover advances in avocado genetics, breeding, and the role of genomic markers.
- There does not appear to be excessive self-citation or citation of irrelevant material.
Tables and Figures
- Figures appear to be central to the results and should be clearly presented with legends.
- Please ensure that all supplementary materials are available and logically structured.
- Consider adding a summary table of key SNP markers, perhaps a supplemental file, noting their potential application.
Author Response
Methods
- The choice and justification for the minimum coverage threshold (4.69×): Why is this optimal?
A mean coverage of 4.69x across samples as part of the implementation of the low-coverage whole genome resequencing (lcWGS) technique is standard and allows maintaining high genotyping quality as well as adequate coverage for SNP detection and phylogenetic inference analysis. For instance, the study conducted by Jiang et al. (2019) observed that coverage levels lower than ~4x result in a significant increase in false positive calls and a reduction in useful genomic coverage; while with an acceptable coverage close to 4.38x, more than 95% of the genome was covered. This new reference now accompanies the statement that this is an acceptable mean coverage for lcWGS at the end of the first paragraph of the Results subsection 2.1 on “lcWGS Recovered more than 64 Millon SNPs”.
Reference: Jiang, Y., Jiang, Y., Wang, S. et al. Optimal sequencing depth design for whole genome re-sequencing in pigs. BMC Bioinformatics 20, 556 (2019). https://doi.org/10.1186/s12859-019-3164-z
- Details on the imputation approach: Which algorithm/software was used for missing data imputation?
Please mind that no imputation of missing data was performed and instead stringent filtering enabled retaining high quality SNP markers across samples.
- Clarify whether samples were balanced across races/geographical clusters. For LMMs, specify the fixed and random effects considered.
Indeed the subpool of samples utilized for racial discrimination was optimized for racial ancestral purity (min. 80% of ancestry to one of the racial clusters). This strategy aimed to maximize discrimination power. The random effect used as part of the LMM was the kinship matrix, as is now specified within the Materials and Methods section (subsection 4.8 on “Identification of SNPs Associated with Avocado Races”).
- Software versions and scripts used for key analyses (phylogeny, ADMIXTURE, SNP filtering) should be referenced.
The phylogenetic tree was originally constructed using FastTree version 2.1.11, and ADMIXTURE version 1.3.0 was used for the analysis. However, the new phylogenetic inference now uses more up to date Bayesian inference as done in MrBayes. These details are now presented in Materials and Methods subsections 4.6 and 4.7 on “Phylogenetic Tree Reconstruction Based on lcWGS-derived SNPs” and “Genetic Structure of Diverse Avocado Tree Samples”, respectively.
- If possible, include a flowchart/table summarizing the overall analysis pipeline for clarity.
The flowchart summarizing the overall analysis pipeline is now presented as Figure S5 for clarity.
Results
- Reporting of SNP numbers, mapping statistics, and quality measures is solid, but make sure all supplementary tables are made available.
Indeed, all supplemental material has been uploaded in the Susy system.
- Phylogenetic clustering and ADMIXTURE results: Figures referenced (e.g., Figure 2) seem crucial but are not included in the provided text. Ensure that all figures are available to reviewers/readers.
All figures are now provided both within the manuscript as well as separate files.
- The explanation of cluster establishment (e.g., K = 5, ADMIXTURE) is clear, but please provide the following in the main text:
8.1. A concise definition of cut-off thresholds for assigning cluster membership (e.g., “>80% ancestry for assignment to a cluster”).
We classified each sample into a specific genetic cluster based on the proportional ancestry inferred from the ADMIXTURE analysis. We used a cut-off threshold of >80% ancestry to assign an individual to a single cluster. Samples with ancestry proportions below this threshold were considered admixed or hybrid individuals and were reported accordingly. This threshold was selected because it provides a balanced trade-off between accurate cluster assignment and retention of genetic variability and has been widely applied in previous studies on perennial crops and tree species (Bouffartigue et al. 2019). This new reference now accompanies the statement on the cut-off threshold of >80% at the end of the first paragraph of the Results subsection 2.3 on “LMMs Recovered 254 Race-Informative SNP Markers” as well as at the end of the first paragraph of the Materials and Methods subsection 4.8 on “Identification of SNPs Associated with Avocado Races”.
Reference: Bouffartigue, C., Debille, S., Fabreguettes, O. et al. (2019). Two main genetic clusters with high admixture between forest and cultivated chestnut (Castanea sativa Mill.) in France. Annals of Forest Science 77, 74 (2020). https://doi.org/10.1007/s13595-020-00982-w
8.2. A short description or legend for all supplementary figures.
Legends for all supplementary figures are enlisted after the legends of the supplementary tables at the end of the subsection “Supplementary Materials” after the section 5 on "Conclusions". Specifically they read as: “Figure S1. Sequencing depth and cumulative sequencing depth. Figure S2. Cross-validation (CV) error from the admixture-based unsupervised genetic clustering (from K = 2 to K = 10) in 205 avocado tree samples with 64,310,961 lcWGS-derived SNP markers. Figure S3. Admixture-based unsupervised genetic clustering (from K = 2 to K = 10) in 205 avocado tree samples with 64,310,961 lcWGS-derived SNP markers. Figure S4. Principal component analysis (PCA) unsupervised genetic clustering with 9,826 high-quality SNP markers in all 205 avocado tree samples, so that racial controls are labeled and colored top down from left to right, as follows: pink for Mexican (ME) race, light blue for Guatemalan (GU) race, light green for West Indian (WI) race, dark blue for Caribbean Colombian (CoCA), dark green for Andean Colombian (CoA), and light gray for avocado samples identified as hybrids between different genetic groups. Figure S5. Flowchart summarizing the overall analysis pipeline (blue boxes refer to input data, green boxes indicate analytical tools, and orange boxes mark obtained/prospective results).”
8.3. Table/Figure clarity: Please make sure that sample color legends (yellow, orange, green, blue, red) are explained in each figure note for improved readability.
As detailed in the legends of Figures 3, 4 and S4 (see comment above), colors are now explained in detail within the figure captions, complementing the racial color assignment accompanying each figure, and matching racial subdivision across figures. Additionally, we have now replaced the maximum likelihood phylogenetic tree in Figure 2 with an updated version (using more up to date Bayesian inference) following the same color convention for racial clustering as in the other figures.
8.4. For reporting the SNP markers (254 candidates), briefly mention the genomic distribution: Are they distributed across races or are some races over/under-represented? Are functional annotations available for these SNPs?
At the end of the Results section, in the subsection 2.3 on “LMMs Recovered 254 Race-Informative SNP Markers”, we now clarify that “Most of these SNPs mapped to non-coding regulatory regions, as expected for putatively neutral, demographic-informative markers, not necessarily in linkage disequilibrium with selection footprints”.
Discussion & Conclusions
- Please discuss potential limitations (e.g., sampling bias, coverage variability, or need for independent validation).
In the Discussion subsection 3.5 on “Perspectives” we now mention more specifically these caveats and offer avenues to bridge them by saying at the beginning of the first paragraph “the identification and comparison of genomic islands of divergence would serve as independent validation to determine if similar genomic regions have been repeatedly recruited during the diversification of the Mexican, Guatemalan, West Indian, and Colombian lineages, or whether different loci underlie their independent evolutionary trajectories”, at the end of the fourth paragraph “after accounting for sampling bias and coverage variability, lcWGS would precisely make possible to carry out high-resolution genome-wide association studies (GWAS), narrowly mapping quantitative trait nucleotide (QTNs) across genetic and environmental background”, and at the beginning of the fifth paragraph “once such candidate trait associated makers are independently validated, they could then be integrated with the 254 SNPs suggested here for racial tracing, enabling indirect mark-er-assisted (MAS) and genomic (GS) selection pipelines ”.
- Expand briefly on how these findings could be incorporated in ongoing breeding or clonal authentication programs.
The second paragraph of the Discussion subsection 3.2 on “Candidate SNPs for Racial Tracing Supports Diversified Avocado Rootstock Production” offers some insight on how these findings could be incorporated in ongoing breeding or clonal authentication programs by saying “the availability of ancestry-informative markers helps overcome one of the persistent challenges in avocado breeding, which is the undocumented origin and high hetero-zygosity of seedling rootstocks, both factors having traditionally bottlenecked se-lection efforts. This panel of candidate SNP markers would enable nurseries to enhance the value of their avocado grafting material by pre-screening seedling rootstocks for racial identity prior to grafting. As result, producers would gain access to a broader and more customized offer of planting material, including: (i) seedling rootstocks with full traceability to the donor tree, (ii) seedling rootstocks pre-labeled by racial background, and (iii) elite clonal rootstocks”.
Additionally, the closing paragraph of the Discussion subsection 3.3 on “Perspectives” also offers some insight on how these findings could be incorporated in ongoing breeding or clonal authentication programs by saying “both approaches could in turn facilitate greater efficiency and precision during early pre-screening of saplings and seedling rootstocks at nursery, not just for racial identity but also for agronomic trait prediction and bioactive compound identification. These efforts will advance the integration of genomics into rootstock selection schemes, while supporting the transition to a more diversified and sustainable avocado production system.”
Tables and Figures
- Figures appear to be central to the results and should be clearly presented with legends.
Now all figures, including the supplemental figures (as is detailed in an aforementioned comment) present detailed captions describing the concordant color codes for racial clustering.
- Please ensure that all supplementary materials are available and logically structured.
All supplementary materials were uploaded in MDPI’s Susy system.
- Consider adding a summary table of key SNP markers, perhaps a supplemental file, noting their potential application.
As the “Data Availability Statement” before the “Acknowledgments” now declares “processed data is contained within the article and as Supplementary Materials. Raw data (205 FASTQ files) is available under NCBI BioProject ID PRJNA1311704”.

Reviewer 2 Report
Comments and Suggestions for Authors
The presented manuscript describes the results of a genomic study of many avocado individuals with an expanded representation of Colombian material. Low-coverage whole genome resequencing (lcWGS) was performed on 410 individuals, including ex situ preserved tree samples, commercial varieties and “criollo” trees from Colombia. On the base of more than 64 millions of SNPs obtained, the authors conducted phylogenetic and population structure analyses. These analyses revealed five genetical clusters (including Mexican, Guatemalan, West Indian races and two clusters from Colombia), as well as many mixed individuals. Using a filtered set of individuals, representing radically pure exemplars of five genetic clusters, the authors identified 254 SNP markers associated with the five avocado genetic races.
The presented article contains new important results that are relevant and in demand. All sections of the article are written clearly and in sufficient detail.
To my opinion, only one part of the article needs correction, exactly phylogenetic analysis. It is conducted and presented insufficiently clear. The following moments rise questions:
1)The authors were intended “to infer the evolutionary relationships among the analyzed avocado samples”. And what was taken as outgroup? For example, in the work by Berdugo-Cely et al. (2023) Persea schiedeana was taken.
2)Bootstrap support for nodes is not shown.
3)Five clusters mentioned in the text are not marked on the phylogenetic tree (Figure 2). It is unclear why these five clusters correspond to three classical botanical races and two Columbian groups. It is better to mark individuals belonging to pure races and groups, as is done in the PCA analysis graphs (Figs. 3 and 5).
4)How did you determine the races: by morphological and physiological features or by another method? This should be described in the Materials and Methods section.
Overall, the article is good, but needs some improvement. So I would recommend a minor revision.
Author Response
- The authors were intended “to infer the evolutionary relationships among the analyzed avocado samples”. And what was taken as outgroup? For example, in the work by Berdugo-Cely et al. (2023) Persea schiedeana was taken.
Thanks for pointing out this key aspect. We have now replaced the maximum likelihood phylogenetic tree in Figure 2 with an updated version this time with rooting (utilizing more up to date Bayesian inference), using as outgroup the Mexican cluster validated as such by Berdugo et al. 2023 (reference already within the manuscript).
- Bootstrap support for nodes is not shown.
As detailed in the previous comment, we have now replaced the maximum likelihood phylogenetic tree in Figure 2 with an updated version, this time using more up to date Bayesian inference and posterior distribution support.
- Five clusters mentioned in the text are not marked on the phylogenetic tree (Figure 2). It is unclear why these five clusters correspond to three classical botanical races and two Columbian groups. It is better to mark individuals belonging to pure races and groups, as is done in the PCA analysis graphs (Figs. 3 and 5).
A very key aspect indeed to maintain cluster consistency across figures. As detailed in the previous two comments, we have now replaced the maximum likelihood phylogenetic tree in Figure 2 with an updated version (using more up to date Bayesian inference), this time following the same color convention for racial clustering as in the other figures.
- How did you determine the races: by morphological and physiological features or by another method? This should be described in the Materials and Methods section.
Racial controls followed the morpho-agronomic classification previously compiled by Berdugo et al. This statement and the corresponding reference are now reiterated at the end of the first paragraph of the Materials and Methods subsection 4.6 on “Phylogenetic Tree Reconstruction Based on lcWGS-derived SNPs”, where we now also explained teh new Bayesian computation.
